# Discovery of the Gene Encoding a Novel Small Serum Protein (SSP) of *Protobothrops flavoviridis* and the Evolution of SSPs

**DOI:** 10.3390/toxins12030177

**Published:** 2020-03-12

**Authors:** Kento Inamaru, Ami Takeuchi, Marie Maeda, Hiroki Shibata, Yasuyuki Fukumaki, Naoko Oda-Ueda, Shosaku Hattori, Motonori Ohno, Takahito Chijiwa

**Affiliations:** 1Department of Applied Life Science, Faculty of Bioscience and Biotechnology, Sojo University, Kumamoto 860–0082, Japan; inashimarushi@gmail.com (K.I.); yuis2.0603@gmail.com (M.M.); mohno218@ybb.ne.jp (M.O.); 2Medical Institute of Bioregulation, Research Center of Genetic Information, Kyushu University, Fukuoka 812–8582, Japan; hshibata@gen.kyushu-u.ac.jp (H.S.); yfukumak@gen.kyushu-u.ac.jp (Y.F.); 3Department of Biochemistry, Faculty of Pharmaceutical Sciences, Sojo University, Kumamoto 860–0082, Japan; naoko@ph.sojo-u.ac.jp; 4Institute of Medical Science, University of Tokyo, Oshima-gun, Kagoshima 894-1531, Japan; shattori@ims.u-tokyo.ac.jp

**Keywords:** small serum proteins, *Protobothrops flavoviridis*, evolution, gene array, comparative genomics

## Abstract

Small serum proteins (SSPs) are low-molecular-weight proteins in snake serum with affinities for various venom proteins. Five SSPs, *Pf*SSP-1 through *Pf*SSP-5, have been reported in *Protobothrops flavoviridis* (“habu”, *Pf*) serum so far. Recently, we reported that the five genes encoding these *Pf*SSPs are arranged in tandem on a single chromosome. However, the physiological functions and evolutionary origins of the five SSPs remain poorly understood. In a detailed analysis of the habu draft genome, we found a gene encoding a novel SSP, SSP-6. Structural analysis of the genes encoding SSPs and their genomic arrangement revealed the following: (1) SSP-6 forms a third SSP subgroup; (2) SSP-5 and SSP-6 were present in all snake genomes before the divergence of non-venomous and venomous snakes, while SSP-4 was acquired only by venomous snakes; (3) the composition of paralogous SSP genes in snake genomes seems to reflect snake habitat differences; and (4) the evolutionary emergence of SSP genes is probably related to the physiological functions of SSPs, with an initial snake repertoire of SSP-6 and SSP-5. SSP-4 and its derivative, SSP-3, as well as SSP-1 and SSP-2, appear to be venom-related and were acquired later.

## 1. Introduction

The bites of viperid snakes, including *Protobothrops flavoviridis* (*Pf)*, cause a variety of symptoms, including bleeding, necrosis, edema, and neurotoxicity, and can be fatal in severe cases. Recent transcriptomic and proteomic studies have identified multiple components of viperid venoms [1,2,3], including phospholipases A_2_ [4,5,6,7], metalloproteases (snake venom metalloproteases, SVMPs) [8,9,10,11], and serine proteases [12,13]. Many of these venom proteins have isoforms. In contrast to neurotoxic group IA PLA_2_s of elapid (Elapinae and Hydrophiinae) venoms, group IIA-PLA_2_s of viperid (Viperinae and Crotalinae) venoms [14], such as hemolytic neutral [Asp^49^]PLA_2_s [15,16], edema-inducing basic [Asp^49^]PLA_2_s [17,18], neurotoxic highly basic [Asp^49^]PLA_2_s [19], and myotoxic [Lys^49^]PLA_2_s [15,20,21,22], diversified via accelerated evolution, in which nucleotide substitutions occurred predominantly at non-synonymous sites.

In contrast, snakes bitten by themselves or other snakes do not show severe symptoms as humans do. Snake serum is able to neutralize or inhibit snake venom activities. Phospholipase A_2_ inhibitors (PLIs) [23,24,25,26,27] and habu serum factor (HSF) [28,29], which inhibit the hemorrhage induced by SVMPs, are able to neutralize these venom activities. Recently, we found a low-molecular-weight serum protein that specifically binds to myotoxic [Lys^49^]PLA_2_ isozymes and revealed that this is a homolog of Small Serum Protein-2 (SSP-2), a human prostatic secretory protein superfamily of 94 amino acids (PSP94) [30]. From *P. flavoviridis* serum, five SSPs, *Pf*SSP-1, *Pf*SSP-2, *Pf*SSP-3, *Pf*SSP-4, and *Pf*SSP-5, have been identified to date [30,31]. However, in terms of blood content, *Pf*SSP-4 and *Pf*SSP-5 are significantly less abundant than *Pf*SSP-1, *Pf*SSP-2, and *Pf*SSP-3 [31]. SSPs are two-domain proteins [31]. The variable N-terminal domains are thought to be involved in binding diverse target molecules, whereas the C-terminal domain, which is largely conserved among the five SSPs, is assumed to be involved in forming oligomers with HSF [32]. *Pf*SSP-2 and *Pf*SSP-5 show high affinity for triflin, a neurotoxin-like protein that blocks muscle contraction [33,34]. *Pf*SSP-1 and *Pf*SSP-4 show affinity for HV1, a low-molecular-weight SVMP that induces apoptosis of vascular endothelial cells [11,35]. *Pf*SSP-3 binds to flavorase, a non-hemorrhagic SVMP [36]. *Pf*SSP-2 binds to [Lys^49^]PLA_2_s [37]. cDNAs encoding five *Pf*SSPs have been isolated and sequenced [30,31]. Interestingly, the cDNAs encoding *Pf*SSP-3 and *Pf*SSP-4 are interrupted by nonsense mutations at the same site on the fourth exon, so as to express truncated mature proteins. The genome fragment containing the genes encoding *Pf*SSP-1 and *Pf*SSP-2 has also been isolated and sequenced [38].

Recently, we revealed that genes for the five PfSSPs, *PfSSP-4*, *PfSSP-5*, *PfSSP-1*, *PfSSP-2*, and *PfSSP-3*, are arranged in tandem in this order on one chromosome of *P. flavoviridis* [39]. According to the configuration of nucleotide sequences in the introns, such as long interspersed nuclear elements (LINEs), DNA transposons, and repetitive sequences, the five *PfSSP*s can be divided into two subgroups: the Long SSP subgroup consists of *PfSSP-1*, *PfSSP-2*, and *PfSSP-5*, and the Short SSP subgroup consists of *PfSSP-3* and *PfSSP-4*. Mathematical analysis of the nucleotide sequences of *PfSSP*s showed that *PfSSP*s in the Short SSP subgroup have evolved in an accelerated manner, whereas those in the Long SSP subgroup have evolved alternately in accelerated and neutral manners. Ortholog analysis of *SSP* genes from five snakes, including non-venomous snakes, suggested that these genes emerged in the order of their configuration on the chromosome. Moreover, a comparison of the arrays of *SSP* genes of five snakes showed that the genome segment encompassing *SSP-1* to *SSP-2* of *Protobothrops* has been inverted. Chromosome inversion appears to have preserved non-synonymous nucleotide substitutions, providing evidence of accelerated evolution [39].

In the present study, we discovered a gene encoding a novel *P. flavoviridis* small serum protein, named *PfSSP-6*, in the 5′ region upstream of the array of five *PfSSP*s [39]. From a detailed structural analysis, we propose: (1) a novel classification of SSPs, (2) an evolutionary scenario to explain SSP paralogs, (3) a relationship between arrays of *SSP* paralogs in the snake genome and environmental conditions, (4) relationships between SSP evolution and physiological functions of their products, suggesting an initial repertoire of *SSP-6*, *SSP-5*, and *SSP-4,* and the subsequent appearance of venom-related *SSP-1, SSP-2*, and *SSP-3*.

## 2. Results and Discussion

### 2.1. Discovery of the Gene Encoding a Novel SSP, *Pf*SSP-6, Far Upstream of the Array of Five *PfSSP* Genes

blastn and tblastx analyses of the habu (HabAm1) database [40] using the nucleotide sequence of *PfSSP-5* as a query revealed that Scaffold 2858 contains an approximately 3.7 kbp sequence similar to the nucleotide sequence of *PfSSP-5*. The amino acid sequence of its N-terminal domain differs from those of the five known *PfSSP*s, whereas its C-terminal domain is very similar. Ten cysteines are conserved among *PfSSP*s. Therefore, this nucleotide sequence was determined to encode a novel type of SSP which was named *PfSSP-6*. To sequence *PfSSP-6*, genomic PCR was performed using the draft nucleotide sequence of the corresponding region in Scaffold 2858 of the Amami Island *P. flavoviridis* genome as a reference. A 1453 bp genome segment that encompassed the 5′ terminus of the putative first exon to the 3′ terminus of the putative third exon of *PfSSP-6*, and another 2375 bp segment that encompassed the 5′ terminal of the putative third exon to 85 bp downstream from the putative fourth exon of *PfSSP-6* were then acquired. Finally, the 3642 bp sequence of *PfSSP-6* was determined. Referring to the construction of *PfSSP-5*, definitive exon–intron boundaries of *PfSSP-6* were identified. *PfSSP-6* consists of four exons and three introns and encodes a 111 amino acid protein, including a 19 amino acid signal peptide. The deduced amino acid sequence of the mature protein encoded by *PfSSP-6* shows 33%–61% identity with the other five PfSSPs, and the positions of its 10 cysteine residues are conserved (Figure 1). Referring to the draft nucleotide sequence of Scaffold 2858 encompassing *PfSSP-6* to *PfSSP-4*, which is the 5′ terminal gene of the array of five *PfSSP*s [39], further genomic PCR with the Amami–Oshima *P. flavoviridis* genome was performed and the 12,406 bp sequence of the intergenic region between *PfSSP-6* and *PfSSP-4*, named *Pf*I-Reg64, was determined (Figure 2).

### 2.2. Sequence Configurations Classify the Six *PfSSP*s Into Three Subgroups

Introns of *PfSSP-6* contained insertions of specific nucleotide sequences, fragments of L1, chicken repeat-1 (CR1), and Gypsy LINEs, fragments of a reverse transcriptase (RT) domain of L2 LINE, fragments of Mariner and hobo-Ac-Tam3 (hAT) DNA transposons, and repetitive sequences, as in the other five *PfSSP*s (Figure 3). The five inserted fragments, L1 and CR1 LINEs in the first intron and Mariner-iii, Gypsy-i, and Gypsy-ii in the third intron, are conserved in all *PfSSP*s. These insertions must therefore have occurred before the divergence of the six *PfSSP*s. Second, configurations of the nucleotide sequences inserted into the second or third intron classified the six *PfSSP*s into two subgroups, Long SSPs and Short SSPs [39]. Long *PfSSP*s are characterized by the fragment of the RT domain of L2 LINE in the third intron. However, the nucleotide sequence of the fragment of the RT domain of L2 LINE in the third intron of *PfSSP-6* differs from those inserted into the other three genes of conventional Long SSPs, *PfSSP-1*, *PfSSP-2*, and *PfSSP-5*. The fragment of L2 LINE in the third intron of three *PfSSP*s, *PfSSP-1*, *PfSSP-2*, and *PfSSP-5*, is truncated in the 3′ terminal region (Figure 1). On the other hand, the fragment of L2 LINE in the third intron of *PfSSP-6* is truncated from the 5′ terminal region, as in typical LINEs [42]. L2 LINE is composed of two open reading frames, ORF1 and ORF2, in which ORF1 encodes an RNA-binding protein and ORF2 encodes a two-domain protein consisting of an endonuclease (EN) and an RT domain [42]. The RT domain of L2 LINE consists of 10 subdomains numbered from zero to IX and a carboxy-terminal conserved region (CTCR) which is thought to serve as the scaffold of reverse-transcription of L2 LINE [43]. A 320 bp section of the L2 LINE fragment in *PfSSP-1* encodes three subdomains, zero to II, of the RT domain. A 431 bp section of that in *PfSSP-2* encodes four subdomains, zero to III, of the RT domain, and 1011 bp of the L2 LINE fragment in *PfSSP-5* encodes nine subdomains, zero to VIII, of the RT domain [39]. However, 1240 bp of the L2 LINE fragment in *PfSSP-6* encode eight subdomains, III to X, and CTCR of the RT domain. This indicates that this L2 LINE is truncated in the 5′ terminal region. It is highly likely that the nucleotide sequence from the 3′ downstream region of the third exon of *PfSSP-6* to the 5′ terminal of the inserted L2 LINE fragment of *PfSSP-6* has disappeared, accompanied by 5′ truncation of L2 LINE. These characteristics indicate that *PfSSP-6* should be classified as a novel Long SSP. Interestingly, body map analysis using semi-quantitative RT-PCR showed that *PfSSP-6* is strongly expressed in the stomach and weakly in the liver (data not shown). It seems that the product of *PfSSP-6* is irrelevant to its role in blood. Thus, the three configurations of inserted nucleotide sequences classify the six *PfSSP*s into three subgroups, conventional and novel Long SSPs, and Short SSPs.

### 2.3. Configurations of *SSP* Paralogs Relevant to Snake Habitat Conditions

Following Chijiwa et al. [39], blastn and tblastx analysis of the draft genomes of seven snakes *Crotalus viridis* (*Cv*, venomous), *Deinagkistrodon acutus* (*Da*, venomous), *Ophiophagus hannah* (*Oh*, venomous), *Python bivittatus* (*Pb*, non-venomous), *Protobothrops mucrosquamatus* (*Pm*, venomous), *Thamnophis sirtalis* (*Ts*, non-venomous), and *Vipera berus* (*Vb*, venomous), in addition to the habu, *P. flavoviridis*, revealed orthologous relationships and configurations of *SSP*s (Figure 4). The genome of the non-venomous Burmese python, *P. bivittatus* (India), contains an ortholog of *PfSSP-6*, named *PbSSP-6*, and three paralogs of *PfSSP-5*, named *PbSSP-5α*, *PbSSP-5β*, and *PbSSP-5γ(Ψ)*. The genome of the garter snake, *T. sirtalis,* a North American colubrid, contains an ortholog of *PfSSP-6*, named *TsSSP-6*, an ortholog of *PfSSP-4*, called *TsSSP-4*, and two paralogs of *PfSSP-5*, *TsSSP-5α*, and *TsSSP-5β*. The genome of the European adder, *V. berus*, a viperid, contains orthologs of *PfSSP-6* and *PfSSP-4* in one scaffold (2247), called *VbSSP-6* and *VbSSP-4* in this study. However, the nucleotide sequences of the second and fourth exons of *VbSSP-4* remain unknown. In addition, the ortholog of *PfSSP-5*, named *VbSSP-5*, was also found in another *V. berus* scaffold (15,659). The genome of the prairie rattlesnake, *C. viridis*, (North America) possesses orthologs of *PfSSP-6*, *PfSSP-4*, and *PfSSP-5* on Chromosome 9, named *CvSSP-6*, *CvSSP-4*, and *CvSSP-5* in this study. The genome of the king cobra, *O. hannah*, (India) has orthologs of *PfSSP-6* and *PfSSP-4*, named *OhSSP-6* and *OhSSP-4(Ψ)*, and three paralogs of *PfSSP-5*, named *OhSSP-5α*, *OhSSP-5β*, and *OhSSP-5γ* in one scaffold (4527). One ortholog and two paralogs of *OhSSP*s were renamed in this study. Two nucleotide segments, previously annotated as *OhSSP-1* and *OhSSP-2* [39], were acquired via genomic PCR to determine their nucleotide sequences. Their nucleotide and deduced amino acid sequences revealed that they are paralogs of *PfSSP-5* and should be renamed *OhSSP-5β* and *OhSSP-5γ*. Therefore, the nucleotide sequence, already annotated as *OhSSP-5(Ψ)* [39], was also renamed *OhSSP-5α*. Structural analysis of the L2 LINE fragment in the third intron showed that *SSP-1*, *SSP-2*, and *SSP-5* belong to the conventional Long SSP subgroup. In addition, the locations of those alleles also suggested that *OhSSP-5β* and *OhSSP-5γ* are evolutionarily related to *SSP-1* and *SSP-2* in *D. acutus*, *P. mucrosquamatus*, and *P. flavoviridis*. The genome of the hundred-pace viper, *D. acutus*, (Southeast Asia) contained orthologs *PfSSP-6*, *PfSSP-4*, *PfSSP-1*, *PfSSP-2*, and *PfSSP-3* in one scaffold (405), named *DaSSP-6*, *DaSSP-4*, *DaSSP-1*, *DaSSP-2*, and *DaSSP-3* in this study. Only the nucleotide segment corresponding to the second exon of the ortholog of *DaSSP-5* was found in the intergenic region between *DaSSP-4* and *DaSSP-1*. Therefore, this fragmented *DaSSP-5* is described as *DaSSP-5δ(Ψ)* in Figure 4. The genome of the Taiwan habu, *P. mucrosquamatus*, (Taiwan) contained an ortholog of *PfSSP-6*, named *PmSSP-6*, and orthologs of *PfSSP-5*, *PfSSP-1*, *PfSSP-2,* and *PfSSP-3* in one scaffold (462), named *PmSSP-5*, *PmSSP-1*, *PmSSP-2*, and *PmSSP-3*. In addition, an ortholog of *PfSSP-4*, named *PmSSP-4*, was also found in another scaffold (21,362). Orthologs of *PfSSP-6* are conserved in the genomes of all eight snakes, whether venomous or non-venomous. Chijiwa et al. showed that *SSP-5* and *SSP-4* were the initial genes in the conventional Long and Short SSP subgroups, respectively [39]. Moreover, the current study revealed that the initial repertoire of SSP genes in the genomes of all snakes should be two genes, encoding SSP-6 for the novel Long SSP subgroup and SSP-5 for the conventional Long SSP subgroup, and that the gene encoding SSP-4 was acquired specifically in the genomes of venomous snakes.

Configurations of *SSP* paralogs in each snake genome were used to classify the eight snakes into three groups. Non-venomous *P. bivittatus* formed the first group, in which two genes encoding SSP-6 and SSP-5 were present in the genome. *T. sirtalis*, *V. berus*, *C. viridis*, and *O. hannah* formed a second group in which three genes encoding SSP-6, SSP-5, and SSP-4 were present. *D. actus*, *P. mucrosquamatus*, and *P. flavoviridis* formed a third group with two genes encoding SSP-1 and SSP-2, in addition to the initial three genes, *SSP-6*, *SSP-4,* and *SSP-5*. This result suggests that the configuration of *SSP* paralogs is relevant to habitat characteristics of each snake. Snakes in the third group, *D. actus*, *P. mucrosquamatus*, and *P. flavoviridis*, inhabit the Orient, where the warm and humid climate might provide richer and more diversified prey than in Europe and America. It is likely that their venom proteins have become varied, and that the serum proteins that neutralize those venoms then also diversified. *O. hannah* did not need to develop novel varieties of IIA-PLA_2_ isozymes; it had another type of venom PLA_2_, the neurotoxic IA-PLA_2_, a lethal component. Therefore, *OhSSP-5β* and *OhSSP-5γ*, corresponding to *SSP-1* and/or *SSP-2*, may have had no need to become derivatives as the counterpart of variable IIA-PLA_2_s.

### 2.4. Diversified *SSP*s Acquired by Advanced Snakes Have More Complex Venom Compositions

Genes encoding SSP paralogs of each snake were analyzed mathematically. The *K*_A_/*K*_S_ ratio, which is the relative ratio of synonymous to nonsynonymous substitutions between the ORFs (Table 1, Table 2, Table 3, Table 4, Table 5, Table 6, Table 7 and Table 8), or *K*_N_, which is the rate of substituted nucleotides between the introns, were calculated (Table 9, Table 10, Table 11, Table 12 and Table 13). However, for genes for which full-length nucleotide sequences of exons or introns remained unknown, the rate of *K*_A_/*K*_S_s for *DaSSP-5δ(Ψ)*, *PbSSP-5γ*, and *VbSSP-4*, or *K*_N_s for *PbSSP*s, *TsSSP*s, and *VbSSP*s were not calculated. In the previous section, we proposed that the initial repertoire of *SSP*s in the genomes of venomous snakes comprised *SSP-6*, *SSP-5*, and *SSP-4*. Our mathematical analysis suggested that these three differ in their characteristics. For any snake, the *K*_A_/*K*_S_ ratio estimated between the ORFs of *SSP-6* and *SSP-5* was the lowest, or considerably lower than the *K*_A_/*K*_S_ ratios estimated between other paralogs. On the other hand, the *K*_A_/*K*_S_ ratios estimated between *SSP-*6 and *SSP-4* and between *SSP-5* and *SSP-4* were close to one. Our interpretation of these results is as follows. SSP-6 or SSP-5 are irrelevant for neutralizing venom proteins and have constitutive or essential roles, such as digestion or blood homeostasis. Therefore, nucleotide sequences of *SSP-6* and *SSP-5* have been conserved. On the other hand, *SSP-4*, acquired in the genomes of venomous snakes, may have encoded the first SSP with a role specific to venom neutralization in the event of accidental bites. Therefore, *SSP-4* may have had to be more plastic than *SSP-5* and *SSP-6*.

The *K*_A_/*K*_S_ ratios estimated between *DaSSP-1* and *DaSSP-2*, *DaSSP-3* and *DaSSP-4*, *PmSSP-1* and *PmSSP-2*, *PmSSP-3* and *PmSSP-4*, *PfSSP-1* and *PfSSP-2*, and *PfSSP-3* and *PfSSP-4* were 1.61, 1.77, 1.49, 1.35, 1.80, and 1.42, respectively (Table 2, Table 5, and Table 6), and the rates of *K*_N_ were 0.0227, 0.005, 0.154, 0.0397, 0.0317, and 0.0283, respectively (Table 10, Table 12, and Table 13). These results showed that the branching of these genes, especially late genes such as *SSP-1*, *SSP-2*, or *SSP-3*, occurred in an accelerated manner, and that the time that passed after their divergence was very short. In addition, the *K*_A_/*K*_S_ ratios estimated between *SSP-1* and *SSP-5* or *SSP-2* and *SSP-5* of *D. acutus*, *P. mucrosquamatus*, and *P. flavoviridis* were around 0.7. This result also supports the idea that *SSP-1* or *SSP-2* and *SSP-5* are evolutionarily related, as suggested above. That is, *SSP-1* and *SSP-2* were recently derived from *SSP-5* and then diversified in an accelerated manner to accommodate venom proteins. SSP-3, the truncated SSP acquired as the successor to SSP-4, is also thought to bind more venom proteins than SSP-4, as do SSP-1 and SSP-2 relative to SSP-5. Therefore, the *K*_A_/*K*_S_ ratios estimated between *SSP-3* and *SSP-4* also show considerably higher values. The many reports that venom proteins bind to SSP-1 [35], SSP-2 [33,37], or SSP-3 [36] also support the above idea. Since animal venoms work as defense mechanisms, as tools to catch prey, or simply to enhance digestion, they should be sensitive to the surrounding environment. Because venom proteins have become more diversified in environments where there are more diverse prey, serum proteins required to neutralize venom activities, such as *SSP-1*, *SSP-2*, and *SSP-3*, have also diversified in an accelerated manner. Even among conventional Long SSPs, *SSP-1*, *SSP-2*, and *SSP-5* have evolved in an accelerated or neutral manner, depending on whether they deal with venom components. On the other hand, *SSP-3* and *SSP-4*, which specifically arose as anti-venom proteins, have evolved in an accelerated manner.

Chijiwa et al. proposed that most nucleotide substitutions at non-synonymous sites occur only immediately after gene duplication. Then random mutations accumulate over time, and selective pressure that leaves “neutral” mutations at synonymous sites erases the traces of accelerated evolution [39]. However, in the genomes of the viperids *P. flavoviridis* and *P. mucrosquamatus*, inversion of the genome segment encompassing *SSP-1* to *SSP-2* occurred, and subsequent accumulation of random mutations was suppressed [39]. These findings are also applicable to *D. acutus DaSSP-1* and *DaSSP-2*. Since the *SSP-3* allele is located in the 3′ region downstream of the inverted genome segment containing *SSP-1* and *SSP-2*, the inversion may also have suppressed accumulation of random mutations in *PfSSP-3*.

## 3. Materials and Methods

### 3.1. Materials

*P. flavoviridis* specimens were provided by the Institute of Medical Sciences of the University of Tokyo. The tail of an *O. hannah* specimen was provided by the Japan Snake Center. That of a *P. mucrosquamatus* was provided by the Medical Institute of Bioregulation, at the Research Center of Genetic Information, Kyushu University. High-molecular-weight genomic DNA was prepared from livers or tails of the snakes according to the method of Blin and Stafford [44]. Total RNA was prepared from various snake organs, according to the ISOGEN protocol (Nippon Gene, Toyama, Japan). Restriction endonucleases and KOD plus DNA polymerase were purchased from Nippon Gene and TOYOBO (Osaka, Japan), respectively. Other reagents and antibiotics were from Nacalai Tesque (Kyoto, Japan) and TAKARA BIO (Shiga, Japan). Specific oligonucleotide primers were synthesized by GENNET (Fukuoka, Japan).

### 3.2. Cloning and Sequencing of the Genome Segment Containing *PfSSP-6*

A dedicated database, HabAm1, [40] was constructed to carry out blastn and tblastx analysis with the nucleotide sequences of *PfSSP*s (*PfSSP-1*–*PfSSP-5*) as queries. Exon–intron boundaries were then determined based on the five *PfSSP*s. Referring to the nucleotide sequence of Scaffold 2858, the sense primer SSP6-5UTR-1, 5′-ggC gTC CCT CCT TCT CCT Tg-3′, which anneals specifically to the first exon of *PfSSP-6*, and the antisense primer SSP6ex3-2, 5′-CTC gCA TTC CAT ACA ATT ggC Tg-3′, which anneals specifically to the third exon of *PfSSP-6*, were used to amplify the 1453 bp genome fragment (Table 14). The sense primer, SSP6ex3-1, 5′-TgT ggC CAA CCA AAT gCg Tgg-3′, which anneals specifically to the third exon of *PfSSP-6*, and the antisense primer SSP6-3flank-1, 5′-CAg CTA TgC ATg CCT TAT ATC AC-3′, which anneals specifically to 85 bp 3′ downstream of the fourth exon of *PfSSP-6*, were then used to amplify the 2363 bp genome fragment (Table 14). Amplified genome fragments were ligated to the pCR™-Blunt II-TOPO^®^ vector (Life Technologies, Carlsbad, CA, USA) and transformed with DH5α-competent cells (TAKARA BIO, Shiga, Japan). Nucleotide sequences were determined using an ABI 3130xl capillary sequencer. The 1453 bp PCR fragment overlapped with the 2363 bp PCR fragment by 89 bp. The physical structure of the 3727 bp segment encompassing the first exon of *PfSSP-6* to 85 bp 3′ downstream of the fourth exon of *PfSSP-6* was determined. This 3727 bp DNA fragment contained four exons encoding *Pf*SSP-6. Moreover, to acquire the nucleotide sequence of the intergenic region between *PfSSP-6* and *PfSSP-4*, named *Pf*I-Reg64, genomic PCR was carried out against the Amami–Oshima *P. flavoviridis* genome and the sense primer Ireg64-1, 5′-CTC CAT gCA AAg gAg gAT TTC C-3′, which anneals to the 3′ terminus of the third intron of *PfSSP-6*, and the antisense primer Ireg64-6, 5′-TAg gCC TTg ACA CAT gAT ggC-3′, which anneals to the middle portion of *Pf*I-Reg64, were used to amplify the 7717 bp genome fragment, named *Pf*IREG64-I (Table 14). The *Pf*IREG64-I fragment was also cloned and sequenced. The 7717 bp *Pf*IREG64-I overlapped with the 3727 bp *PfSSP-6* by 474 bp. The sense primer Ireg64-5, 5′-CAT TgT TgA gCA ACC CTT ggC-3′, which anneals 2501 bp 5′ upstream of Ireg64-6, and the antisense primer Ireg64-8 5′-ggA CTA TTA AgC AgT ggA ATg gC-3′, which anneals 2340 bp 5′ upstream of the first exon of *PfSSP-4* (3′ terminal of *Pf*IReg-64), were then used to amplify the 5283 bp genome fragment, named *Pf*IREG64-II (Table 14). The *Pf*IREG64-II fragment was also cloned and sequenced. The 5283 bp *Pf*IREG64-II overlapped with the 7717 bp *Pf*IREG64-I by 2523 bp. The sense primer Ireg64-9, 5′-ggC CCT CTT CCA Agg ACA AgC-3′, which anneals 455 bp 5′ upstream of Ireg64-8, and the antisense primer Ireg64-10, 5′-ACC TCg TTC CTC CAg CCA CT-3′, which anneals to the 5′ terminus of the first intron of *PfSSP-4*, were then used the 2971 bp genome fragment, named *Pf*IREG64-III (Table 14). The *Pf*IREG64-III fragment was also cloned and sequenced. The 2971 bp *Pf*IREG64-III overlapped with the 5267 bp *Pf*IREG64-II by 455 bp. Finally, the physical structure of the 16,248 bp segment encompassing the third intron of *PfSSP-6* to the first intron of *PfSSP-4* was completely established. The nucleotide sequences of *PfSSP-6* and the genome segment from *PfSSP-6* to *PfSSP-4* are available from the Genbank/EMBL/DDBJ databases under Accession No. LC518073.

### 3.3. RepeatMasker Analysis of the Nucleotide Sequence of *PfSSP-6*

A dedicated database was constructed with repetitive sequences of the genomes of various organisms collected from Repbase of the Genetic Information Research Institute [45]. RepeatMasker utilized the nucleotide sequences of *PfSSP-6* against the database via BLAST+, RMBlast (NCBI), and Tandem Repeats Finder (Boston University) [46].

### 3.4. Determination of Nucleotide Sequences and Chromosomal Configurations of Genes Encoding Orthologs of *PfSSP*s from Seven Snake Taxa

Draft nucleotide sequences of the genomes of seven taxa, *C. viridis* (*Cv*, venomous) [47], *D. acutus* (*Da*, venomous) [48], *O. hannah* (*Oh*, venomous) [49], *P. bivittatus* (*Pb*, non-venomous) [50], *P. mucrosquamatus* (*Pm*, venomous) [51], *T. sirtalis* (*Ts*, non-venomous) [52], and *V. berus* (*Vb*, venomous) [53], were downloaded to create a dedicated genome database. Referring to the nucleotide and amino acid sequences of *PfSSP*s deduced using tblastn or blastn, the nucleotide sequences encoding orthologs of *PfSSP*s and their flanking regions in each snake genome were ascertained. The *T. sirtalis* genome segment containing *TsSSP-4* and *TsSSP-5α* and the three *P. bivittatus* genes, *PbSSP-5α*, *PbSSP-5β*, and *PbSSP-5γ (Ψ)*, were identified in separate scaffolds; therefore, their locations and arrangements are tentative.

### 3.5. Determining the Nucleotide Sequences of *SSP* Paralogs of *P. mucrosquamatus* and *O. hannah*

To acquire complete nucleotide sequences of *PmSSP-3*, *PmSSP-4*, *OhSSP-1*, *OhSSP-2*, *OhSSP-5,* and *OhSSP-6*, genomic PCR was performed on the *O. hannah* and *P. mucrosquamatus* genomes to amplify two overlapping nucleotide segments separately. These included the 5′ segment of the gene encompassing the first exon to the second exon, and the 3′ segment of the gene encompassing the second exon to the fourth exon of each gene.

For *PmSSP-3* (*Pm* Scaffold 462), the sense primer, PmSSP34-5UTR, 5′-CAA ggg TTg gTC TTg gTT TTT g-3′, which anneals to the 5′ terminus of the first exon of *PmSSP-3* and *PmSSP-4*, and the antisense primer, PmSSP3ex2-R, 5′-ggT AgA gAA AAg CCC CCA AAg-3′, which anneals to the second exon of *PmSSP-3*, were used to amplify the 1169 bp 5′ segment of *PmSSP-3* (Table 15). The sense primer, PmSSP3-F, 5′-TgC TTT ggg ggC TTT TCT C-3′, which anneals to the middle portion of the second exon of *PmSSP-3*, and the antisense primer, PmSSP34-R, 5′-CTT gAC TgA GAC TgA AgT TCC-3′, which anneals to the 311 bp 3′ region downstream of the fourth exon of *PmSSP-3* and *PmSSP-4*, were then used to amplify the 2722 bp 3′ segment of *PmSSP-3* (Table 15). The 5′ segment of *PmSSP-3* overlapped with the 3′ segment of *PmSSP-3* by 31 bp. The physical structure of the 3860 bp segment encompassing the first exon of *PmSSP-3* to the 311 bp at the 3′ region downstream of the fourth exon of *PmSSP-3* was completed.

With regard to *PmSSP-4*, the sense primer, PmSSP34-5UTR, described above, and the antisense primer PmSSP4ex2-R, 5′-CgT TTC Agg TAA Agg AAT ACT C-3′, which anneals to the second exon of *PmSSP-4* based on the nucleotide sequence of *Pm* Scaffold 21,362, were used to amplify the 1139 bp 5′ portion of *PmSSP-4* (Table 15). Using *Pm* Scaffold 21,362, the sense primer, PmSSP4-F, 5′-gAg TAT TCC TTT ACC TgA AAC g-3′, which anneals to the middle portion of the second exon of *PmSSP-4*, and the antisense primer, PmSSP34-R, described above, were then used to amplify the 2999 bp 3′ segment of *PmSSP-4*. The 5′ segment of *PmSSP-4* overlapped with the 3′ segment of *PmSSP-4* by 22 bp (Table 15). The physical structure of the 4118 bp segment encompassing the first exon of *PmSSP-4* to the 311 bp at the 3′ region downstream of the fourth exon of *PmSSP-4* was sequenced.

For *OhSSP-5α* (*Oh* Scaffold 4527), the sense primer, OhSSPs-5UTR, 5′-ATA AAT Tgg Agg AgC RgA TTC CT-3′, which anneals to the common nucleotide sequence of the 5’ UTR of *OhSSP*s, and the antisense primer, OhSSP5-ex2-R, 5′-CTC AgC TTC AAA gCC CCA gg-3′, which anneals to the second exon of *OhSSP-5*, were used to amplify the 1107 bp 5′ segment of *OhSSP-5α* (Table 15). The sense primer OhSSP5-F, 5′-gAg CAT gCT TTA CCT ggg gC-3′, which anneals to the middle portion of the second exon of *OhSSP-5α* (*Oh* Scaffold 4527), and the antisense primer, OhSSP5-R 5′-TCC ATg TgT AgA gAT CAA ACA Cg-3′, which anneals to the middle portion of the fourth exon of *OhSSP-5α* (*Oh* Scaffold 47,978), were the used to amplify the 2130 bp 3′ part of *OhSSP-5α* (Table 15). The 5′ segment of *OhSSP-5α* overlapped with the 3′ segment of *OhSSP-5α* by 32 bp. The sequence of the 3205 bp segment encompassing the first exon of *OhSSP-5α* to the fourth exon of *OhSPP-5α* was determined.

In regard to *OhSSP-5β*, the sense primer, OhSSPs-5UTR, described above, and the antisense primer, OhSSP2-ex2-R, 5′-CTC AgC TTC AAA gAg CCC TCT-3′, which anneals to the second exon of *OhSSP-5β* (*Oh* Scaffold 4527), were used to amplify the 2456 bp 5′ section of *OhSSP-5β* (Table 15). The sense primer, OhSSP2-F, 5′-gAg CAT gCT ATA gAg ggC TCT-3′, which anneals to the middle portion of the second exon of *OhSSP-5β*, and the antisense primer, OhSSP2-R, 5′-gAT CAA ACA TCA CAg CgC TgC-3′, which anneals to the fourth exon of *OhSSP-5β*, were then used to amplify the 2180 bp 3′ section of *OhSSP-5β* (Table 15). The 5′ segment of *OhSSP-5β* overlapped with 3′ segment of *OhSSP-5β* by 42 bp. The 4594 bp segment encompassing the first exon of *OhSSP-5β* to the fourth exon of *OhSPP-5β* was sequenced.

For *OhSSP-5γ*, the sense primer, OhSSPs-5UTR, described above, and the antisense primer, OhSSP1-ex2-R, 5′-TTA Agg AAC ACT CCA AAg CAC C-3′, which anneals to the second exon of *OhSSP-5γ* (*Oh* Scaffold 4527), were used to amplify the 1534 bp 5′ segment of *OhSSP-5γ* (Table 15). The sense primer, OhSSP1-F, 5′-gAg ggT gCT TTg gAg TgT TCC-3′, which anneals to the middle portion of the second exon of *OhSSP-5γ* (*Oh* Scaffold 4527), and the antisense primer, OhSSP1-R, 5′-gAT CAg ACA CCA CAg CTg Tgg-3′, which anneals to the fourth exon of *OhSSP-5γ*, were then used to amplify the 1999 bp 3′ segment of *OhSSP-5γ*. The 5′ half segment of *OhSSP-5γ* overlapped with the 3′ half segment of *OhSSP-5γ* by 26 bp (Table 15). The structure of the 3508 bp segment encompassing the first exon of *OhSSP-5γ* to the fourth exon of *OhSPP-5γ* was deciphered.

For *OhSSP-6*, the sense primer, OhSSPs-5UTR, described above, and the antisense primer, OhSSP6-ex2-R, 5′-TAA ACT gAg gTT TAA AgA gAT CCA-3′, which anneals to the second exon of *OhSSP-6* (*Oh* Scaffold 10,541), were used to amplify the 1734 bp 5′ segment of *OhSSP-6* (Table 15). The sense primer, OhSSP6-F, 5′-gCA gCA TgC TTC ATg gAT CTC-3′, which anneals to the middle portion of the second exon of *OhSSP-6* (*Oh* Scaffold 10,541), and the antisense primer, OhSSP6-R, 5′-CCg TgT gAA AAg NTC AgA CAT C-3′, which anneals to the fourth exon of *OhSSP-6* (*Oh* Scaffold 12,359), were then used to amplify the 3461 bp 3′ segment of *OhSSP-6* (Table 15). The 5′ section of *OhSSP-6* overlapped with the 3′ segment of *OhSSP-6* by 43 bp. The sequence of the 5152 bp segment encompassing the first exon of *OhSSP-6* to the fourth exon of *OhSPP-6* was determined. Nucleotide sequences of *OhSSP-6*, *OhSSP-5α*, *OhSSP-5β*, *OhSSP-5γ*, *PmSSP-4*, and *PmSSP-3* are available in the Genbank/EMBL/DDBJ databases under Accession Nos. LC518074–LC518078 and LC519888.

### 3.6. Expression Analysis of *PfSSP-6* mRNA Using Semi-Quantitative RT-PCR

First-strand cDNAs from snake organs were synthesized by reverse transcription and primer extension with a SMART cDNA Library Construction Kit (Clontech, California, USA). Based on nucleotide sequences of the genes encoding *PfSSP-6*, the sense primer SSP6-5UTR-1, described above, and the antisense primer SSP6-3UTR-2, 5′- ACA TgA gAg ATT TAT TCC AgT gTg -3′, which anneals to the 3′ terminal of the fourth exon of *PfSSP-6*, were designed (Table 14). cDNA of β-actin, designated as ACTB, was amplified as an internal standard with the sense primer, SHU7, 5′-CAg AgC AAg AgA ggT ATC CN-3′ (N = G, A, T, C), and the antisense primer, SHU8, 5′-TAg ATg ggC ACA gTg Tgg gN-3′, as described previously [54].

### 3.7. Mathematical Analysis

Alignment of the amino acid sequences of snake SSPs was performed using ClustalX software. Nucleotide sequences of ORFs encoding the mature SSPs were rearranged and gaps in the aligned amino acid sequences were removed using PAL2NAL. The rates of synonymous (*K*_S_) and nonsynonymous (*K*_A_) substitutions per site between the ORFs of the genes were calculated using the Nei–Gojobori method, as implemented in PAML [55]. After removing LINEs, DNA transposons, and indels (insertion/deletion) from the introns, alignment of introns was performed using ClustalX. Values of *K*_N_ that estimated rates of substituted nucleotides between the introns of *SSP*s were calculated from the aligned sequence data.

## Figures and Tables

**Figure 1 toxins-12-00177-f001:**
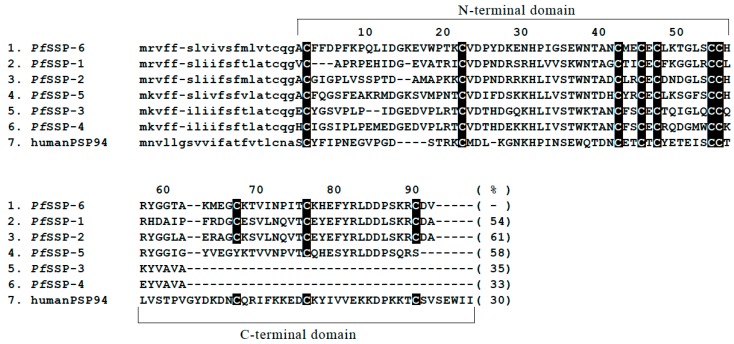
Alignment of the deduced amino acid sequences encoded by the open reading frames (ORFs) of six *PfSSP*s and human *PSP94*. Position numbers refer to amino acid residues of the mature proteins. Signal peptide sequences are in lower case letters. The cysteines are shaded. Abbreviations: *Pf*, *P. flavoviridis*. References: *PfSSP-6* (this study); *PfSSP-1* (AB360906.1); *PfSSP-2* (AB360907.1); *PfSSP-5* (AB360910.1); *PfSSP-3* (AB360908.1); *PfSSP-4* (AB360909.1) [31]; human *PSP94* (NP_002434.1) [41]. Numerals in parentheses show percent identities of *Pf*SSPs and human PSP94 with *Pf*SSP-6.

**Figure 2 toxins-12-00177-f002:**
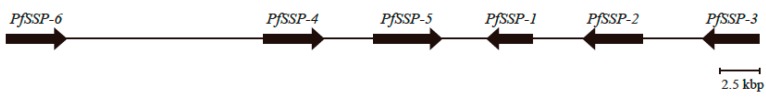
Schematic representation of the 16,048 bp genome segment containing *PfSSP-6* in the 5′ region upstream of the array of five *PfSSP*s. Bold arrows indicate the areas and transcription directions of the genes in the segment.

**Figure 3 toxins-12-00177-f003:**
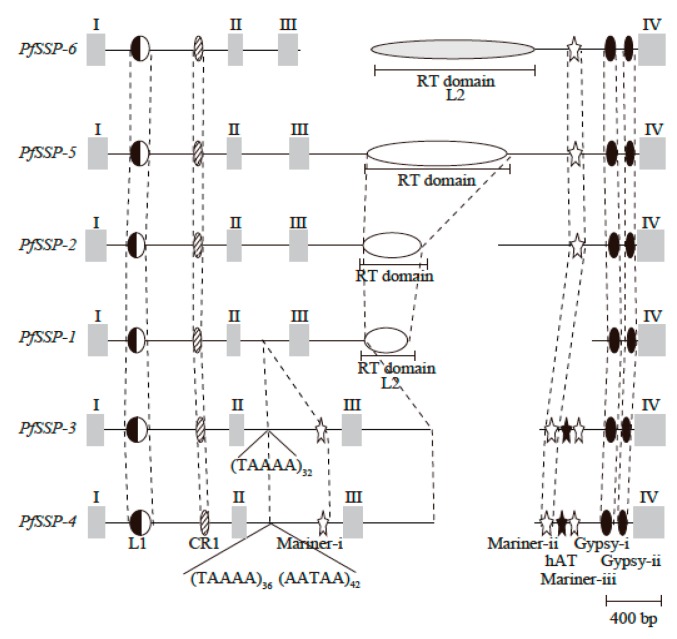
Schematic configurations of nucleotide sequences in the introns of six *PfSSP*s. Gray bars represent exons. Half-closed, hatched, open, and closed ellipses represent fragments of L1, CR1, L2, and Gypsy LINEs, respectively. Since the fragment of L2 LINE in the third intron of *PfSSP-6* differs from those of L2 LINE in the third introns of *PfSSP-1*, *2*, and *5*, the ellipse of *PfSSP-6* is shown in gray. Open and closed stars represent the fragments of Mariner and hAT DNA transposons. Positions of the corresponding fragments are linked with dashed lines. Positions of repetitive sequences (TAAAA and AATAA) are indicated with carets and numbers of repetitions are indicated as subscripts.

**Figure 4 toxins-12-00177-f004:**
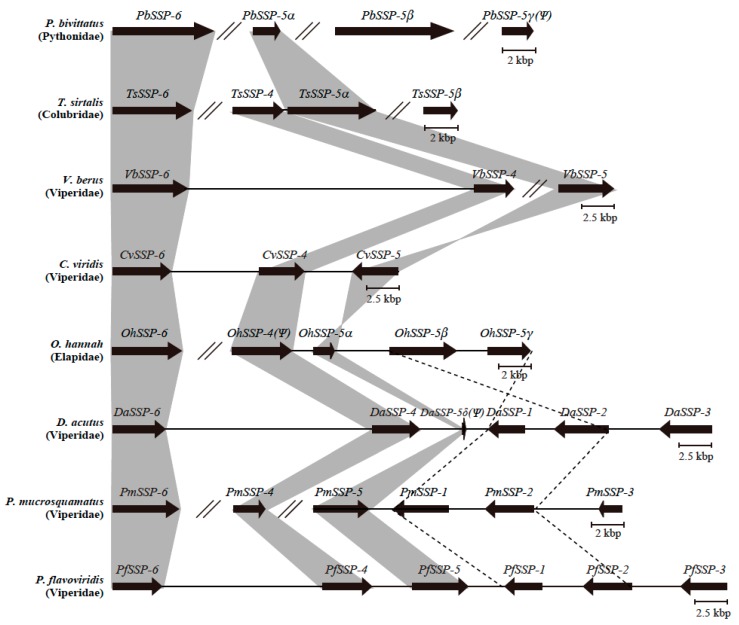
Schematic comparison of the arrays of *SSP*s of eight snake taxa. Abbreviations: *Cv*: *C. viridis*; *Da*: *D. acutus*; *Oh*: *O. hannah*; *Pb*: *P. bivittatus*; *Pf*: *P. flavoviridis*; *Pm*: *P. mucrosquamatus*; *Ts*: *T. sirtalis*; *Vb*: *V. berus*. Bold arrows indicate the areas and transcription directions of the genes. Orthologs of *SSP-6*, *SSP-5*, and *SSP-4* in each snake genome are linked with gray. Inverted genome segments of *O. hannah*, *D. acutus*, *P. mucrosquamatus*, and *P. flavoviridis* are linked with dashed lines. Double slashes indicate interruptions of the nucleotide sequences.

**Table 1 toxins-12-00177-t001:** *K*_A_/*K*_S_ ratios estimated between the ORFs of *C. viridis SSP*s.

	*CvSSP-4*	*CvSSP-5*	*CvSSP-6*
*CvSSP-4*		0.749	0.879
*CvSSP-5*			0.306
*CvSSP-6*			

**Table 2 toxins-12-00177-t002:** *K*_A_/*K*_S_ ratios estimated between the ORFs of *D. acutus SSP*s.

	*DaSSP-1*	*DaSSP-2*	*DaSSP-3*	*DaSSP-4*	*DaSSP-6*
*DaSSP-1*		1.61	0.934	0.823	0.832
*DaSSP-2*			0.849	0.705	1.03
*DaSSP-3*				1.77	0.878
*DaSSP-4*					0.830
*DaSSP-6*					

**Table 3 toxins-12-00177-t003:** *K*_A_/*K*_S_ ratios estimated between the ORFs of *O. hannah SSP*s.

	*OhSSP-4* *(* *Ψ* *)*	*OhSSP-5* *α*	*OhSSP-5* *β*	*OhSSP-* *5γ*	*OhSSP-* *6*
*OhSSP-4* *(* *Ψ* *)*		0.931	0.875	0.700	0.919
*OhSSP-5* *α*			0.836	0.719	0.666
*OhSSP-5* *β*				0.832	0.685
*OhSSP-* *5γ*					0.509
*OhSSP-* *6*					

**Table 4 toxins-12-00177-t004:** *K*_A_/*K*_S_ ratios estimated between the ORFs of *P. bivittatus SSP*s.

	*PfSSP-* *5* *α*	*PfSSP-* *5β*	*PbSSP-6*
*PbSSP-* *5* *α*		1.594	0.343
*PbSSP-* *5β*			0.296
*PbSSP-6*			

**Table 5 toxins-12-00177-t005:** *K*_A_/*K*_S_ ratios estimated between the ORFs of *P. mucrosquamatus SSP*s.

	*PmSSP-1*	*PmSSP-2*	*PmSSP-3*	*PmSSP-4*	*PmSSP-5*	*PmSSP-6*
*PmSSP-1*		1.49	0.891	0.821	0.639	0.694
*PmSSP-2*			0.825	0.662	0.476	0.620
*PmSSP-3*				1.35	0.479	0.780
*PmSSP-4*					0.586	0.889
*PmSSP-5*						0.370
*PmSSP-6*						

**Table 6 toxins-12-00177-t006:** *K*_A_/*K*_S_ ratios estimated between the ORFs of *P. flavoviridis SSP*s.

	*PfSSP-1*	*PfSSP-2*	*PfSSP-3*	*PfSSP-4*	*PfSSP-5*	*PfSSP-6*
*PfSSP-1*		1.80	0.660	0.790	0.597	0.792
*PfSSP-2*			0.808	0.781	0.504	0.891
*PfSSP-3*				1.40	0.599	0.990
*PfSSP-4*					0.670	1.07
*PfSSP-5*						0.626
*PfSSP-6*						

**Table 7 toxins-12-00177-t007:** *K*_A_/*K*_S_ ratios estimated between the ORFs of *T. sirtalis SSP*s.

	*TsSSP-4*	*TsSSP-5* *α*	*TsSSP-* *5* *β*	*TsSSP-6*
*TsSSP-4*		1.06	0.603	0.651
*TsSSP-5* *α*			0.675	0.447
*TsSSP-* *5β*				0.659
*TsSSP-6*				

**Table 8 toxins-12-00177-t008:** *K*_A_/*K*_S_ ratios estimated between the ORFs of *V. berus SSP*s.

	*VbSSP-5*	*VbSSP-6*
*VbSSP-5*		0.604
*VbSSP-6*		

**Table 9 toxins-12-00177-t009:** *K*_N_ values estimated between the introns of *C. viridis SSP*s.

	*CvSSP-4*	*CvSSP-5*	*CvSSP-6*
*CvSSP-4*		0.319	0.358
*CvSSP-5*			0.372
*CvSSP-6*			

**Table 10 toxins-12-00177-t010:** *K*_N_ values estimated between the introns of *D. acutus SSP*s.

	*DaSSP-1*	*DaSSP-2*	*DaSSP-3*	*DaSSP-4*	*DaSSP-6*
*DaSSP-1*		0.0227	0.248	0.253	0.249
*DaSSP-2*			0.247	0.251	0.246
*DaSSP-3*				0.0050	0.285
*DaSSP-4*					0.288
*DaSSP-6*					

**Table 11 toxins-12-00177-t011:** *K*_N_ values estimated between the introns of *O. hannah SSP*s.

	*OhSSP-4(Ψ)*	*OhSSP-5α*	*OhSSP-5β*	*OhSSP-5γ*	*OhSSP-6*
*OhSSP-4(Ψ)*		0.331	0.340	0.324	0.530
*OhSSP-5α*			0.0615	0.0801	0.277
*OhSSP-5β*				0.0857	0.226
*OhSSP-5γ*					0.275
*OhSSP-6*					

**Table 12 toxins-12-00177-t012:** *K*_N_ values estimated between the introns of *P. mucrosquamatus SSP*s.

	*PmSSP-1*	*PmSSP-2*	*PmSSP-3*	*PmSSP-4*	*PmSSP-5*	*PmSSP-6*
*PmSSP-1*		0.154	0.338	0.342	0.339	0.374
*PmSSP-2*			0.281	0.284	0.262	0.316
*PmSSP-3*				0.0397	0.293	0.346
*PmSSP-4*					0.296	0.349
*PmSSP-5*						0.295
*PmSSP-6*						

**Table 13 toxins-12-00177-t013:** *K*_N_ values estimated between the introns of *P. flavoviridis SSP*s.

	*PfSSP-1*	*PfSSP-2*	*PfSSP-3*	*PfSSP-4*	*PfSSP-5*	*PfSSP-6*
*PfSSP-1*		0.0317	0.251	0.258	0.248	0.231
*PfSSP-2*			0.254	0.261	0.253	0.229
*PfSSP-3*				0.0283	0.261	0.279
*PfSSP-4*					0.270	0.287
*PfSSP-5*						0.267
*PfSSP-6*						

**Table 14 toxins-12-00177-t014:** Primers used to acquire the nucleotide sequences from the genome domain encompassing *PfSSP-6* to *PfSSP-4*. The symbols (f) or (r) after the position numbers indicate the directions of the primers. Forward or reverse denote whether the direction of elongation was the same or opposite to that of transcription. Nucleotide positions refer to nucleotide sequences reported in this study (LC518073).

Name	Positions	Nucleotide Sequence (GC Content: %, Tm: °C)
SSP6-5UTR-1	1–20 (f)	5′- ggC gTC CCT CCT TCT CCT Tg -3′ (65, 66)
SSP6ex3-2	1431–1453 (r)	5′- CTC gCA TTC CAT ACA ATT ggC Tg-3′ (48, 68)
SSP6ex3-1	1365–1385 (f)	5′- TgT ggC CAA CCA AAT gCg Tgg -3′ (57, 66)
SSP6-3UTR-2	3581–3604 (r)	5′- ACA TgA gAg ATT TAT TCC AgT gTg-3′(38, 66)
SSP6-3flank-1	3705–3727 (r)	5′- CAg CTA TgC ATg CCT TAT ATC AC -3′ (43, 66)
Ireg64-1	3254–3275 (f)	5′- CTC CAT gCA AAg gAg gAT TTC C -3′ (50, 66)
Ireg64-6	10,950–10,970 (r)	5′- TAg gCC TTg ACA CAT gAT ggC -3′ (52, 64)
Ireg64-5	8449–8469 (f)	5′- CAT TgT TgA gCA ACC CTT ggC -3′ (52, 64)
Ireg64-8	13,709–13,731 (r)	5′- ggA CTA TTA AgC AgT ggA ATg gC -3′ (48, 68)
Ireg64-9	13,277–13,297 (f)	5′- ggC CCT CTT CCA Agg ACA AgC -3′ (62, 68)
Ireg64-10	16,228–16,247 (r)	5′- ACC TCg TTC CTC CAg CCA CT -3′ (60, 64)

**Table 15 toxins-12-00177-t015:** Primers utilized to determine nucleotide sequences of genome fragments containing *SSP*s of *P. mucrosquamatus* and *O. hannah*. The symbols (f) or (r) after the position numbers indicate the directions of the primers. Forward or reverse indicate whether the direction of elongation was the same or opposite to that of transcription. Nucleotide positions refer to the nucleotide sequences reported in this study. Abbreviations: *Oh: O. hannah*; *Pm*: *P. mucrosquamatus*.

Name	Scaffold	Nucleotide Sequence (GC Content: %, Tm: °C)
PmSSP34-5UTR (f)	*Pm* Scaffold 462	5′- CAA ggg TTg gTC TTg gTT TTT g -3′ (45, 64)
PmSSP3ex2-R (r)	*Pm* Scaffold 462	5′- ggT AgA gAA AAg CCC CCA AAg -3′ (52, 64)
PmSSP3-F (f)	*Pm* Scaffold 462	5′- TgC TTT ggg ggC TTT TCT C -3′ (47, 56)
PmSSP34-R (r)	*Pm* Scaffold 462	5′- CTT gAC TgA GAC TgA AgT TCC -3′ (45, 62)
PmSSP4ex2-R (r)	*Pm* Scaffold 462	5′- CgT TTC Agg TAA Agg AAT ACT C -3′ (41, 62)
PmSSP4-F (f)	*Pm* Scaffold 21,362	5′- gAg TAT TCC TTT ACC TgA AAC g -3′ (41, 62)
OhSSPs-5UTR (f)	*Oh* Scaffold 4527	5′- ATA AAT Tgg Agg AgC RgA TTC CT -3′ (43, 66)
OhSSP5-ex2-R (r)	*Oh* Scaffold 4527	5′- CTC AgC TTC AAA gCC CCA gg -3′ (60, 64)
OhSSP5-F (f)	*Oh* Scaffold 4527	5′- gAg CAT gCT TTA CCT ggg gC -3′ (60, 64)
OhSSP5-R (r)	*Oh* Scaffold 47,978	5′- TCC ATg TgT AgA gAT CAA ACA Cg -3′ (43, 66)
OhSSP2-ex2-R (r)	*Oh* Scaffold 4527	5′- CTC AgC TTC AAA gAg CCC TCT -3′ (52, 64)
OhSSP2-F (f)	*Oh* Scaffold 4527	5′- gAg CAT gCT ATA gAg ggC TCT -3′ (52, 64)
OhSSP2-R (r)	*Oh* Scaffold 4527	5′- gAT CAA ACA TCA CAg CgC TgC -3′ (52, 64)
OhSSP1-ex2-R (r)	*Oh* Scaffold 4527	5′- TTA Agg AAC ACT CCA AAg CAC C -3′ (52, 64)
OhSSP1-F (f)	*Oh* Scaffold 4527	5′- gAg ggT gCT TTg gAg TgT TCC -3′ (45, 64)
OhSSP1-R (r)	*Oh* Scaffold 4527	5′- gAT CAg ACA CCA CAg CTg Tgg -3′ (57, 66)
OhSSP6-ex2-R (r)	*Oh* Scaffold 10,541	5′- TAA ACT gAg gTT TAA AgA gAT CCA -3′ (33, 64)
OhSSP6-F (f)	*Oh* Scaffold 10,541	5′- gCA gCA TgC TTC ATg gAT CTC -3′ (52, 64)
OhSSP6-R (r)	*Oh* Scaffold 12,359	5′- CCg TgT gAA AAg NTC AgA CAT C -3′ (50, 66)

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
