# Peer review of "Discovery of the Gene Encoding a Novel Small Serum Protein (SSP) of Protobothrops flavoviridis and the Evolution of SSPs"

_toxins, 2020, doi:10.3390/toxins12030177_

Round 1

Reviewer 1 Report

Dear Authors,

the manuscript you proposed has several merits. They include a genomic dissection of the mentioned locus as well evolutionary analyses to better describe homologues of such family. However my major concers are on the form used to present it.

The overall scientific quality of the used methods  and obtained results are evident. Conversely issues arise in the presentation of these data; the sentences resulted fragmented and they appear a list of characters unrelated to each other.

Thus in my opinion tha manuscript should be rewritten and resubmitted.

Author Response

We utilized the MDPI’s English editing services to improve the “fragmented sentences". 

Reviewer 2 Report

Overall, I enjoyed reading results and methods sections of this manuscript, but as it stands I feel the introduction and abstract need to be revised before it reaches the standard for publication in Toxins. The topic of the study is of great interest to the snake venom community, but the study overall suffers from a very narrow focus on only the species at hand. I feel that with a wider focus in the abstract and introduction this manuscript fits within the scope of the journal and will make an excellent contribution to the toxinology literature. My main concern is that from the introduction and abstract I cannot understand what the motivation for the project is. Major comments: Currently, the abstract does not capture the study well or place it context. This is currently the weakest part of the paper and needs to be rewritten. A take home message from the abstract would also improve it. The introduction sets a very narrow scene for the paper and I do not know what the motivation for the project is. The field of toxinology has exploded over the last few years and much of that literature could be included to make the paper broader and more applicable to wider audience and also set the scene for why the study was conducted. This needs to be rectified throughout the paper. Incorporation of recent literature will help both readers and reviewers better understand the importance of your study and how it drives the field forward. The methods and results were both strong sections of the paper. However, I am not sure what the first paragraph in the methods section adds and I suggest you delete it.

Author Response

Abstract has been rewritten to clarify the purpose of this study. Introduction started with a description of more general phenomena and cited new references to make broader readers interested. The results have also been rewritten as concisely as possible. As the first section of Materials and Methods, 3.1 Materials, describes the materials and their sources needed to perform this study, it does not need to be deleted. English editing by MDPI's service was also finished.

Round 2

Reviewer 1 Report

As reported in my first response the soundness of manuscript is ok.

Also the form was modified especially in the introduction, while in other part of manuscript no significant changes were introduced.

good work

Reviewer 2 Report

The authors have done a great job of responding to my comments and I am happy with the current version.